# Impact of the COVID-19 Pandemic and Governmental Policies on Rehabilitation Services and Physical Medicine in Jordan: A Retrospective Study

**DOI:** 10.3390/ijerph20031972

**Published:** 2023-01-20

**Authors:** Nihad A. Almasri, Carl J. Dunst, Hikmat Hadoush, Jaber Aldaod, Yousef Khader, Ali Alrjoub, Ahmad Almasri

**Affiliations:** 1Department of Physiotherapy, School of Rehabilitation, The University of Jordan, Amman 11942, Jordan; 2Orelena Hawks Puckett Institute, Asheville, NC 28730, USA; 3Department of Rehabilitation Sciences, Faculty of Applied Medical Sciences, Jordan University of Science and Technology, Irbid 22110, Jordan; 4Ministry of Health, Amman 11118, Jordan; 5Department of Public Health, Community Medicine, Faculty of Medicine, Jordan University of Science and Technology, Irbid 22110, Jordan

**Keywords:** disability, healthcare, policy, rehabilitation services

## Abstract

*Background*: Although the COVID-19 pandemic led to a series of governmental policies and regulations around the world, the effect of these policies on access to and provision of rehabilitation services has not been examined, especially in low and middle- income countries. *Aims*: The aim of this study was to investigate the impact of governmental policies and procedures on the number of patients who accessed rehabilitation services in the public sector in Jordan during the pandemic and to examine the combined effect of sociodemographic factors (age and gender) and the governmental procedures on this number of patients. *Methods*: A retrospective cohort study was conducted based on records of 32,503 patients who visited the rehabilitation center between January 2020 and February 2021. Interrupted time-series analysis was conducted with three periods and by age and gender. *Results*: The number of patients who visited the rehabilitation clinics decreased significantly between January 2020 and May 2020 due to government-imposed policies, then increased significantly until peaking in September 2020 (*p* = 0.0002). Thereafter, the number of patients decreased between October 2020 and February 2021 as a result of the second wave of the COVID-19 pandemic (*p* = 0.02). The numbers of male and female patients did not differ (*p* > 0.05). There were more patients aged 20 years and older attending rehabilitation clinics than younger patients during the first strict lock down and the following reduction of restriction procedures periods (*p* < 0.05). *Conclusions*: The COVID-19 public measures in Jordan reduced access to rehabilitation services. New approaches to building resilience and access to rehabilitation during public health emergencies are needed. A further examination of strategies and new approaches to building resilience and increasing access to rehabilitation during public health emergencies is warranted.

## 1. Introduction

In March 2020, the outbreak of coronavirus disease (COVID-19) caused the sixth public health emergency of international concern declared by the World Health Organization (WHO). In many countries where rehabilitation services are recognized as an essential part of quality healthcare systems, the COVID-19 pandemic led to increased demands for acute care rehabilitation, with a greater number of rehabilitation professionals being reassigned to emergency rooms and joining the acute care teams in working with hospitalized patients with COVID-19. Although several studies include descriptions of the impact of COVID-19 on rehabilitation services and activities in Europe [1], East Asia [2], and North America [3], studies from the Middle East—and, specifically, Jordan—are lacking.

Decisions to shift, transform, delay, or discontinue rehabilitation services are complex and have likely impacted the quality of life of many patients who needed rehabilitation [2]. Worldwide, PwDs with or without COVID-19 infection were among the worst affected by the reorganization of health services [4]. In Italy, telerehabilitation was proposed as a model for the provision of rehabilitation services; in particular, when the need for rehabilitation services increased as the pandemic progressed [1]. In China, online rehabilitation counseling was utilized to reduce the flow of patients in the rehabilitation department; in addition, rehabilitation services were not offered in acute care settings for patients with COVID-19. In some cases, rehabilitation services were provided in isolated rooms or through phones, videos, and manuals [2]. A study that included 12 countries across Europe, North and South America, and China reported the suspension of outpatient and home-based rehabilitation services and the use of different types of telerehabilitation, such as phones, videos, and virtual rehabilitation [3]. A common theme across the studies was the lack of national plans for rehabilitation services during the pandemic. There was a lack of understanding of the role of rehabilitation in emergencies, highlighting the need for a better understanding of the role of rehabilitation in emergencies.

After the first case of COVID-19 was diagnosed in early March 2020, the Jordanian government imposed a policy of extensive contact tracing followed by quarantine or hospital isolation of symptomatic cases. This was supported by a strict countrywide lockdown, nightly curfew, and border closure on 14 March 2020. Afterward, a Royal Decree was issued approving the implementation of the National Defense Law on 17 March 2020. As of 11 April 2021, there had been 662,395 confirmed cases and 7708 confirmed deaths in Jordan. Two spikes were observed in November 2020 [5] and March 2021. In Jordan, the pandemic had a huge impact on the entire healthcare system, including the public, private, international, and charitable sectors. The emergency departments, intensive care units, laboratory services, and imaging services experienced high demand as the pandemic progressed [1,6].

However, rehabilitation services in Jordan were deemed nonessential during the emergency response. Since its recognition in the 1970s, rehabilitation services have been provided within a traditional medical model [7,8]. Rehabilitation services in public hospitals are limited to outpatient clinics, which are supervised by physical medicine specialists. Acute care rehabilitation services are not available as needed since the outbreak of the pandemic. This situation has had a significant negative impact on patients with the highest risk of deterioration of their functional abilities who were already suffering from significant limitations on participation, such as people with disabilities (PwDs) and the elderly [1]. Patients with acute neuro-motor impairments, such as acute cases of stroke, severe cases of neuropathies, and many other neurological conditions, have not received needed rehabilitation services [6].

It is known that the role of rehabilitation extends to the post-acute phase, during which time patients need services that aim to improve gas exchanges, reduce dyspnea, improve muscle functioning, reduce disability, and optimize function [9,10]. This implies the need for reorganization of the rehabilitation services to meet post-COVID-19 demands, including the need for services that target post-COVID-19 problems. Such problems include, but are not limited to, respiratory, cognitive, central, and peripheral nervous system problems; deconditioning; critical illness-related myopathy and neuropathy; dysphagia; joint stiffness; pain; and psychiatric problems [9]. Therefore, there is a need to implement new rehabilitation strategies and treatments to meet the needs of COVID-19 patients during the acute, post-acute, and chronic stages.

In line with the WHO initiative “Rehabilitation 2030: A Call for Action”, the Cochrane Rehabilitation and the World Health Organization Rehabilitation Programme has published the Rehabilitation Research Framework for COVID-19 patients (CRRF) [11]. The CRRF offers a comprehensive view of the current evidence about the rehabilitation of patients with COVID-19 and highlights under-investigated areas where further studies are needed to inform best practices and ensure high-quality rehabilitation services. The CRRF revealed that most studies focus on the micro-level (individual level) and epidemiological aspects of the pandemic. However, studies related to the meso- and macro-levels, such as access to and regulation of rehabilitation services and changes in policies due to the pandemic, are needed [11]. The aims of this study were to examine (a) the impact of governmental policies and procedures on the number of patients who accessed rehabilitation services in the public sector in Jordan between January 2020 and February 2021, (b) the combined effect of sociodemographic factors (age and gender) and the governmental procedures on the number of patients who visited a rehabilitation center between January 2020 and February 2021, and (c) the most commonly provided rehabilitation services during the pandemic.

## 2. Methods

### 2.1. Ethical Approval and Design

The study was conducted and results were reported following the Strengthening the Reporting of Observational Studies in Epidemiology (STROBE) collaboration guidelines [12]. The study was performed following the principles of the declaration of the World Medical Association Declaration of Helsinki, which was developed by the World Medical Association to ensure the ethical integrity of studies. The Institutional Review Board of the Jordanian Ministry of Health exempted the study from IRB approval because the records provided for the investigators were unidentifiable and patients’ identifying information was not shared.

### 2.2. Settings

The Ministry of Health (MoH) is considered the largest provider of rehabilitation services in Jordan, which comprises three zones: the northern, central, and southern parts of Jordan. Rehabilitation services are provided by rehabilitation teams in 21 public hospitals and centers. There are no recognized acute care rehabilitation services in the hospitals of the MoH, but a rehabilitation team, including a physiatrist, physiotherapist, and other members as needed, provides rehabilitation assessment and intervention based on regular visits to different wards and in response to consultations with other specialties. The Rehabilitation Medicine Department at Al-Bashir hospital is the largest and most comprehensive provider of rehabilitation services in Jordan. As of 2021, the Al-Bashir Hospital included eight Physical Medicine and Rehabilitation Specialists in the MoH hospitals, 45 physiotherapists, 3 occupational therapists, 1 speech and language pathologist, and 12 prosthetics and orthotics among the total staff in the MoH hospitals. Since its establishment in 1965, the rehabilitation services at Al-Bashir hospital have been expanded to provide a comprehensive range of such services, including physiotherapy, occupational therapy, speech and language therapy, orthosis, and prosthesis, as well as physiatrists’ procedural interventions through the clinic for botulinum toxin injection and spasticity treatment, a stroke clinic, a nerve conduction study, a pain management clinic, and a joint clinic for neurosurgery and pediatrics.

### 2.3. Participants

In this retrospective cohort study, we included patients who received rehabilitation services in the largest public hospital in Jordan between January 2020 and February 2021. The medical records of 32,503 patients were reviewed. Patients’ ranged from 1 month old infants to elderly people more than 60 years of age (48% were males and 52% were females). Around 22% of the patients were younger than 20 years of age and 78% were between 20 years and 60+ years. The records were extracted from the national electronic medical system (HAKEEM).

### 2.4. Data Extraction and Variables

The records of all patients who utilized the rehabilitation services at Al-Basheer hospital were extracted from the national electronic medical system (HAKEEM). HAKEEM is linked with patients’ national identification numbers and all the information related to the patients’ healthcare in the public sector was accessed through this electronic database. Retrieved data included the numbers of patients who visited the rehabilitation clinics grouped by age and gender. The data extraction process for this study was initiated by a request from the head of the Physical Medicine Department to have the required unidentified data exported to Excel files by an information technology employee who had no connection to the research team. The researchers did not have access to the full database or patients’ records in HAKEEM. Therefore, the data were anonymous and had no personal identifying information linked to patients’ identities, ensuring the integrity of the data. The study was approved by the Ministry of Health national information board, exempting the study from obtaining patients’ consent.

The independent variables included in this study were three time periods selected based on the governmental policies imposed and participants’ age and gender at the time of their visit to the rehabilitation center. The dependent variable was the number of patients who visited the rehabilitation clinics between January 2020 and February 2021. In this study, the number of patients who visited the rehabilitation clinics during each time period were examined.

### 2.5. Data Analyses

Data were entered into a Microsoft Excel database, where line graphs were used to observe the trends in the number of patients who visited the rehabilitation clinics between January 2020 and February 2021 (dependent variable). To achieve the first aim of the study, we used an interrupted time-series analysis to assess the effects of the governmental lockdown procedures on the number of patients who visited the rehabilitation center in the largest public hospital in Jordan during three time periods: (1) period 1 represented the first wave of the pandemic and the first strict lockdown, which extended from January 2020 until May 2020; (2) period 2 represented the reduction in restriction procedures and extended from June 2020 until September 2020; and (3) period 3 represented the pandemic’s second wave and the second period of strict lockdown and extended from October 2020 until February 2021. Interrupted time-series analysis enabled us to examine changes in the trend lines by calculating the differences between the slopes of the three time periods. The slopes, *p*-value, and standardized coefficients are reported for the differences between the three periods of interest.

To achieve the second aim of the study the combined effect of sociodemographic factors (age and gender) and the governmental procedures on the number of patients who visited the rehabilitation center from January 2020 to February 2021 was examined. An interrupted time-series analysis was also used to assess the effects of the governmental lockdown procedures on the number of patients who visited the rehabilitation center as grouped by gender (male vs. female) and age group (<20 years vs. ≥20 years) during each time period.

To achieve the third aim of the study, descriptive analysis and line graphs were utilized to describe the services provided in the rehabilitation department during the three time periods of the study. The significance level was set at *p* < 0.05. Analyses were conducted using both Excel and SPSS version 25 software.

## 3. Results

### 3.1. Differences among Trend Lines for the Three Periods of Time

The number of patients who visited the rehabilitation clinics decreased sharply between January 2020 and May 2020 (period 1). Only 78 patients accessed the rehabilitation clinics in May 2020 compared to 3394 patients in January 2020 (see Figure 1). Afterward, the number of patients increased until it peaked in September 2020, at which point it exceeded the number of patients before COVID-19. The number of visits declined and stabilized during the period between October 2020 and February 2021 (period 3) corresponding to the governmental policies that included the nationwide lockdown announced on 16 October 2020.

Table 1 shows the results of the comparisons between the trend lines for the numbers of patients seen during the three different time periods. The results indicated a significant difference between the trends for the numbers of patients seen during periods 1 and 2 (b = 71.32, *t* (5) = 10.18, *p* = 0.0002) and during periods 2 and 3 (b = −25.80, *t* (5) = −3.12, *p* = 0.02).

### 3.2. Differences between Trends by Gender

Figure 2 shows the trend lines for the numbers of male and female patients seen between January 2020 and February 2021. Table 2 shows the results of the comparisons between the trend lines for the numbers of male and female patients seen during each of the three time periods. We found a statistically significant difference in the number of female patients versus male patients in favor of females only during period 1 (b = −3.70, *t* (4) = −6.80, *p* = 0.0024), whereas no significant differences were found between the number of male and female patients during periods 2 and 3. This indicated that females accessed rehabilitation services more during the early months of the pandemic and up to the first strict lockdown in the country when the numbers dropped both for males and females. However, without controlling for socioeconomic variables, it is impossible to confirm that this difference is gender-related.

### 3.3. Differences between Trends by Age

Figure 3 shows the trend lines for the number of patients who visited the rehabilitation clinics between January 2020 and February 2021 distinguished into two age groups. Table 3 shows the results of the comparisons between the trend lines for the numbers of patients <20 years of age and ≥20 years of age seen in each of the three different periods. We found statistically significant differences between the numbers of patients who were younger than 20 years of age and those who were older than 20 years of age during period 1 (b = −22.72, *t* (4) = −16.87, *p* = 0.0001) and period 2 (b = 22.73, *t* (4) = 3.98, *p* = 0.0073), whereas no significant difference was found during period 3. It has been observed that less patients under the age of 20 visited the governmental rehabilitation centers both during the strict lockdown as well as after the process of social distancing was eased by the government.

### 3.4. Description of Types of Services

Figure 4 shows the types of services accessed during the three time periods. Almost the same trends were seen for patients accessing specific services. Exercises and prostheses were the two commonly provided services during period 1, the Botox and cerebral palsy clinics were visited the most during period 2, and, finally, electrotherapy and hydrotherapy services were commonly provided during period 3.

## 4. Discussion

In response to the research gaps identified in the Rehabilitation Research Framework for COVID-19 patients [11], this study utilized national data from Jordan’s public health system to examine changes in the numbers of patients who visited rehabilitation clinics during the COVID-19 pandemic mapped onto the governmental procedures; in particular, the country lockdown. The results revealed a significant descending trend in the number of patients from when the pandemic started in March 2020 until May 2020, when the country was in the first strict lockdown. A discernable drop in the number of patients occurred, which reached almost zero in May 2020 during the complete nationwide lockdown, indicating a general underutilization of the rehabilitation services by patients during the first few months of the pandemic. However, a significant increase in the number of patients was noted afterward, reaching a peak in September 2020 after the lockdown procedures were eased across the entire country. A third significant descending trend was noted after September 2020, which plateaued at a low number of patients until February 2021. This changing pattern highlights the impact of the country-level policies and procedures on access to rehabilitation services. Moreover, the numbers of patients significantly differed based on age from January 2020 to February 2021.

At the mesosystem level, the descending trend in the number of patients receiving rehabilitation services in the public health sector in Jordan indicated limited access to public outpatient rehabilitation services during the pandemic. This is consistent with findings from an international study that included 38 countries that showed that up to two million patients did not have access to various rehabilitation services in different settings (acute, post-acute, and outpatient rehabilitation due to cessation of admission to rehabilitation and early discharge) during the COVID-19 pandemic [13]. Prvu Bettger et al. [3] described adjustments to the continuum of rehabilitation services across 12 low-income, middle-income, and high-income countries in the national context of COVID-19, with most of the countries reporting complete suspension of outpatient rehabilitation services and reductions in the provision of inpatient rehabilitation because of shifting of personnel or beds to support the management of COVID-19 patients. In addition, the fact that inpatient (acute care) rehabilitation services do not exist in the public sector in Jordan explains the decrease in the number of patients receiving acute rehabilitation services during the pandemic [14].

At the macrosystem level, demand for rehabilitation services in different settings in Jordan is expected to increase dramatically in the future among patients who did not receive rehabilitation services during the lockdown and post-COVID-19 patients. The overall declining trend in the number of patients accessing rehabilitation services was found to be similar for all patients, regardless of their age. There was, however, a greater number of adult and elderly patients visiting governmental rehabilitation services during the pandemic than younger patients. This can be attributed to the increased need for rehabilitation services among older people during the pandemic. Nevertheless, considering that many young people in Jordan who receive rehabilitation services in the public sector have disabilities [8], further research is needed to examine whether young people were able to access non-governmental rehabilitation services dur-ing that time. It is encouraged that policy makers in Jordan include people with disabili-ties in national response policies in order to ensure that they receive the necessary services [15] 

The pandemic revealed a need for a paradigm shift in the provision of services for children with disabilities all around the world. Based on expert opinion, investigators in two studies recommended that special attention be given to the needs of children with cerebral palsy and autism spectrum disorders and their families during the COVID-19 pandemic; in particular, during periods of quarantine [16,17]. Children with disabilities are a vulnerable population who can be impacted by the pandemic much more than adults and the elderly. The quarantine procedures and the lack of access to rehabilitation services most likely affected the levels of mental and psychological stress among parents, leading to mental health issues [16,17]. Although the results of this study include limited evidence about meeting the needs of children with disabilities during the pandemic, the overall decline in rehabilitation services in medical settings is concerning. Therefore, it is suggested that current models of rehabilitation be considered wherein the role of the family in providing care for their children is enhanced such as family-centered and collaborative models of rehabilitation, particularly for follow-up and continuity of care [18].

Interestingly, a high number of patients were referred by physiatrists to electro- and hydrotherapy clinics in this study. After the lockdown was eased in August 2020, the highest numbers of patients were referred to the Botox and cerebral palsy clinics. This reflects a traditional medical model of services, in which the focus of rehabilitation services is on medical management and modalities rather than functional training and exercises [19]. Evidence-based rehabilitation services are needed in Jordan to adapt to changes in rehabilitation services during and after the pandemic. Jordan’s recently approved national rehabilitation strategy is expected to lead to this change in the public healthcare sector.

### 4.1. Study Limitations

The limitations of this study were mainly due to the retrospective cohort design and the utilization of secondary data, which limited our ability to examine many variables that could have affected the provision of rehabilitation services during the pandemic. For example, we were not able to assess the clinical implications of the reduction in the number of patients receiving rehabilitation during the COVID-19 outbreak due to the lack of clinical outcomes documented in the HAKEEM system. We could not differentiate between regular patients and new patients who accessed rehabilitation during the study timeframe. In addition, since we were unable to access socioeconomic variables in the HAKEEM system, we were not able to control for factors such as income, education, and residency when examining the change in the number of patients who visited the rehabilitation clinics.

### 4.2. Policy Implications

Following the COVID-19 pandemic, demand for rehabilitation services is expected to increase, which will require an increase in supply through health planning and adjustments to the financing of healthcare. It is therefore possible to adopt the following recommendations to help Jordan’s health systems implement resilient rehabilitation services: (1) incorporating acute care rehabilitation services into national health care plans as an essential component of quality care, (2) establishing policies to ensure continuity of rehabilitation during periods of limited access, such as policies regulating telerehabilitation, home programs, and in-home virtual reality services, (3) requiring rehabilitation professionals to attend continuing education courses that address the acute care competencies of the rehabilitation workforce, (4) providing services to people with disabilities through client-centered models, and (5) promoting public-private partnerships among healthcare sectors in order to meet the in-creased demand for rehabilitation services.

## 5. Conclusions

The COVID-19 pandemic public measures in Jordan reduced access to rehabilitation services. A further examination of strategies and new approaches to building resilience and in-creasing access to rehabilitation during public health emergencies is warranted. As part of efforts to limit major disruptions to Jordan’s health care system, current rehabilitation practice models, such as client-centered care models, may be considered. Through these models, patients can actively participate in their rehabilitation process in addition to im-proving their ability to access and navigate the rehabilitation process.

## Figures and Tables

**Figure 1 ijerph-20-01972-f001:**
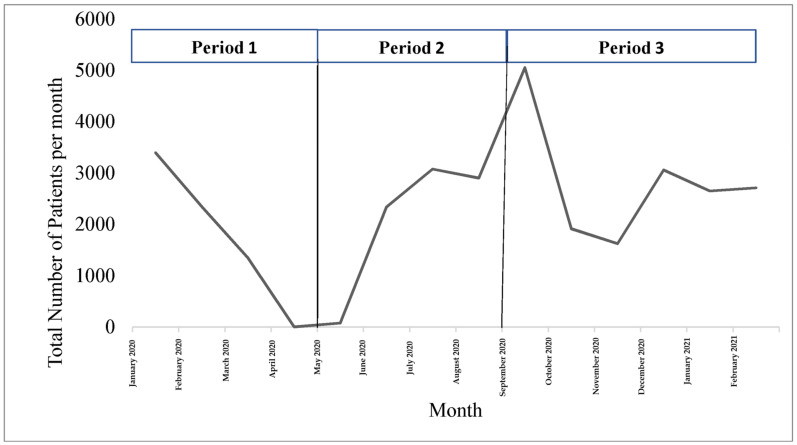
Trends in the number of patients served between January 2020 and February 2021. Period 1: first strict lockdown; Period 2: reduction of restriction procedures; Period 3: second strict lockdown.

**Figure 2 ijerph-20-01972-f002:**
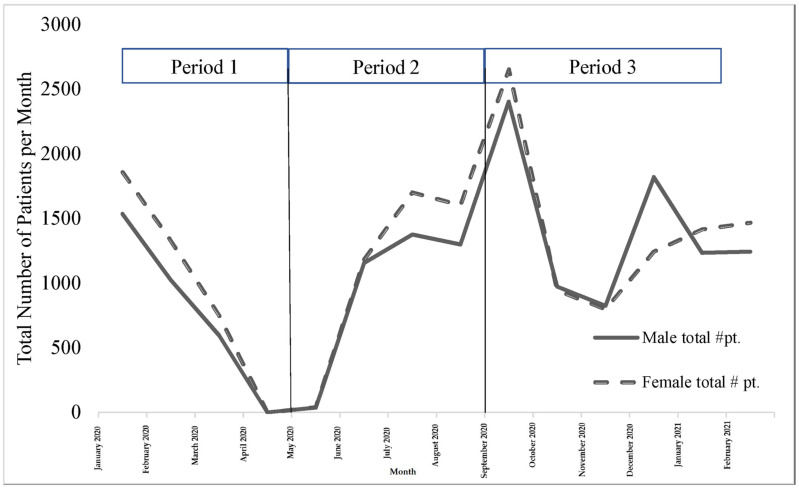
Trends in the numbers of patients seen between January 2020 and February 2021 distinguished by gender. Period 1: first strict lockdown; Period 2: reduction of restriction procedures; Period 3: second strict lockdown.

**Figure 3 ijerph-20-01972-f003:**
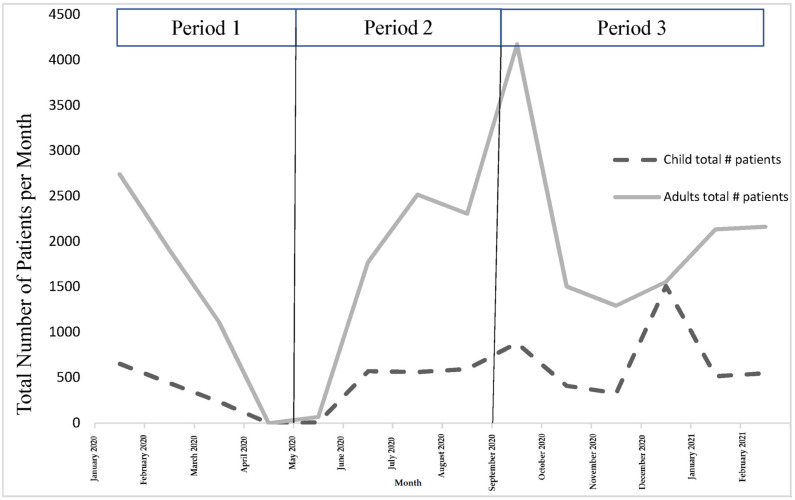
Trends for number of patients seen between January 2020 and February 2021 distinguished by age group. Period 1: first strict lockdown; Period 2: reduction of restriction procedures; Period 3: second strict lockdown.

**Figure 4 ijerph-20-01972-f004:**
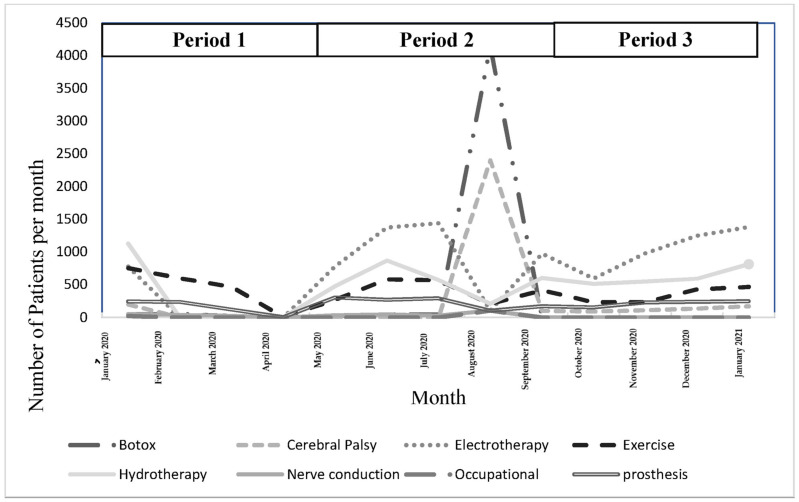
Type of services provided during the study period. Period 1: first strict lockdown; Period 2: reduction of restriction procedures; Period 3: second strict lockdown.

**Table 1 ijerph-20-01972-t001:** Comparisons of the trend lines for the numbers of patients seen during the three time periods.

Time Period	b Coefficient Slope	Standardized β Coefficient	SE of Slope	Between-Trend Differences	*t*-Value	*p*
Period 1	−37.05	−0.998	1.45	71.32	10.18	0.0002
*Jan* *uary* *2020–April 2020*						
Period 2	34.27	0.933	6.85			
*May 2020–September 2020*				−25.80	−3.12	0.0206
Period 3	8.47	0.687	4.63			
*October 2020–February 2021*						

**Table 2 ijerph-20-01972-t002:** Comparison of the trend lines for the numbers of male versus female patients seen during each of the three time periods.

Time Period	Gender	b Coefficient Slope	Standardized β Coefficient	SE of Slope	Between-Slope Differences	*t*-Value	*p*
Period 1 *January 2020–April 2020*	Male	−16.68	−0.999	0.54	−3.70	−6.80	0.0024
	Female	−20.38	−0.997	0.03			
Period 2 *May 2020–September 2020*	Male	15.89	0.918	3.55	2.49	0.51	0.6289
	Female	18.38	0.942	3.37			
Period 3 *October 2020–February 2021*	8.47	3.06	0.391	3.72	2.35	0.59	0.5759
	Female	5.41	0.897	1.37			

**Table 3 ijerph-20-01972-t003:** Comparison of the trends for the numbers of patients visiting rehabilitation services distinguished by age group during each period.

Time Period	Age Group	β Coefficient Slope	Standardized β Coefficient	SE of Slope	Difference	*t*-Value	*p*
Period 1 *January 2020–April 2020*	≤20 years	−7.17	−0.9997	0.11	−22.72	−16.87	0.0001
	>20 years	−29.89	−0.997	1.34			
Period 2 *May 2020–September 2020*	≤20 years	5.77	0.884	1.58	22.73	3.98	0.0073
	>20 years	28.50	0.937	5.49			
Period 3 *October 2020–February 2021*	≤20 years	1.46	0.147	5.05	5.55	1.01	0.3497
	>20 years	7.01	0.863	2.11			

## Data Availability

Not available.

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
