# Peer review of "Impact of the COVID-19 Pandemic and Governmental Policies on Rehabilitation Services and Physical Medicine in Jordan: A Retrospective Study"

_ijerph, 2023, doi:10.3390/ijerph20031972_

Round 1

Reviewer 1 Report

The authors present the importance of rehabilitation services and the association between sociodemographic factors, governmental policies during the pandemic, and access to rehabilitation services.

Several minor suggestions for making the paper better organized:

1. The introduction section needs to be better structured. You should move the description from line 37 to line 54 to the end of the introduction section. Start with describing the situation in other countries (line 87 etc.), then move forward to policies in Jordan. Describing Al-Bashir should be in the method section, not the introduction section. 

2. Please describe the sample of participants (the data set) in the method section.

3. Did you analyze the data using only Excell or other software?  

Author Response

Authors’ response to reviewer 1

Thank you for your comments which improved our paper. Changes are highlighted in yellow in the attached file. Below are the responses to your comments.

Comment 1

The introduction section needs to be better structured. You should move the description from line 37 to line 54 to the end of the introduction section. Start with describing the situation in other countries (line 87 etc.), then move forward to policies in Jordan. Describing Al-Bashir should be in the method section, not the introduction section. 

Authors’ response: We moved lines 37-54 towards the end and moved the lines 87 – end of the paragraph up. The description of Albasheer hospital is moved to methods under the heading settings.

Comment 2  

Please describe the sample of participants (the data set) in the method section.

Authors’ response: We added a participants section to describe the participants. Unfortunately, the database we could access doesn’t include socioeconomic data to better describe the sample mentioned in the study's limitations.

Comment 3

Did you analyze the data using only Excel or other software?  

Authors' response:  We used both Excel and SPSS. 25 and added this to the data analysis section.

Reviewer 2 Report

This is an interesting article on the consequences of the COVID-19 pandemic and government response on rehabilitation services and physical medicine in Jordan. The article is timely and relevant for IJERPH.

The introduction needs some revision. There should be a first paragraph that contextualises the study into a global picture: impact of COVID-19 response on rehabilitation services and beyond. My suggestion is to keep the first sentence as it is, then move up the sentence in lines 66-70 (page 2) and the information about lacking studies in the middle east (line 88-89, page 2), as well as the sentence in lines 82-86 (page 2). Use these elements to write a new introductory paragraph that explains the importance of studying the impact (of COVID-19 on rehabilitation/other services) and in general terms what the ambition of this article is (what is it about).

It is also suggested to move up the paragraph that starts at the bottom of page 2 (lines 87 – 101, without the information about lacking studies in middle east) – and to use this as a second paragraph following the new introduction.

Further about the introduction: On page 3, there is need for an additional sentence or two between the sentence on lines 119-121 to improve the flow – explain what the study is about (in more detail than in the first paragraph).

Ethics: Provide reference number to the ethical approval, either in text or as instructed by the journal.

Data analyses (should be plural in heading, line 159): Although probably simple, include also information on how the third research question/aim will be achieved.

Figure 2. I suggest to include also "Female total" (dotted line) together (over or under) with "Male total".

Discussion: Parts of the text in lines 276-314 (page 9) overlaps with the text under "policy recommendations" and could be shortened. Also, the use of imperative form (needs to, should include) and recommendations fits better in the conclusion rather than in the discussion.

Author Response

Authors’ response to reviewer 2

Thank you for your comments which improved our paper. Changes are highlighted in yellow in the attached file. Below are the responses to your comments.

Comment 1

The introduction needs some revision. There should be a first paragraph that contextualizes the study into a global picture: the impact of COVID-19 response on rehabilitation services and beyond. My suggestion is to keep the first sentence as it is, then move up the sentence in lines 66-70 (page 2) and the information about lacking studies in the middle east (lines 88-89, page 2), as well as the sentence in lines 82-86 (page 2). Use these elements to write a new introductory paragraph that explains the importance of studying the impact (of COVID-19 on rehabilitation/other services) and in general terms what the ambition of this article is (what is it about). It is also suggested to move up the paragraph that starts at the bottom of page 2 (lines 87 – 101, without the information about lacking studies in the middle east) – and to use this as a second paragraph following the new introduction. Further about the introduction: On page 3, there is a need for an additional sentence or two between the sentence on lines 119-121 to improve the flow – explain what the study is about (in more detail than in the first paragraph).

Author’s response: The introduction was revised as requested. The first paragraph now provides the global context of the study and the impact of COVID-19 on rehabilitation services. The lines you referred to are now forming the first paragraph. Then we moved the paragraph at bottom of page 2 to be the second paragraph.  An additional sentence was added to connect lines 121 and 122. That focuses on what is the study aiming to accomplish.

Comment 2 Ethics: Provide a reference number to the ethical approval, either in text or as instructed by the journal.

Authors’ response: The IRB exempted the study from approval because records shared with investigators were unidentifiable. We added this to the ethical approval section.

Comment 3

Data analyses (should be plural in heading, line 159): Although probably simple, include also information on how the third research question/aim will be achieved.

Authors’ response: We changed the heading to be plural. We added information about the third aim of data analysis.

Comment 4

Figure 2. I suggest to include also "Female total" (dotted line) together (over or under) with "Male total".

Authors’ response:  We added the female total to the legend.

Comment 5

Discussion: Parts of the text in lines 276-314 (page 9) overlaps with the text under "policy recommendations" and could be shortened. Also, the use of imperative form (needs to, should include) and recommendations fits better in the conclusion rather than in the discussion.

Authors’ response:  We shortened the redundant parts and moved the recommendation to Policy implication.

Reviewer 3 Report

Thank you for the opportunity to review this interesting manuscript.

The authors present a retrospective analysis of client presentations in their facility during 3 waves of the Covid pandemic in Jordan. They compare the rate of change presentations between waves, between women and men and two main age strata. They draw conclusions about who was affected most by covid public health measures.

Overall the work is clearly presented. There are opportunities to shorten the background, discussion and conclusion. Results are presented clearly, with areas for improvement. My main concerns with the paper are about the interpretations of findings, and the policy recommendations and overall conclusions. These are reflected in line-by-line assessment below.

I extent my regards to the authors and the editors of the journal.

Abstract

Middle and low resource settings is an atypical phrase. Is middle and low income setting or country better? This matters in terms of both consistency and for future searches.

Why are policies and procedures ‘meso’. Isn’t policy macro, while procedures might be a mix of levels? Why is the term needed at all?

18. Delete full stop after patients

25. Is this analysis age corrected? Why is it important that there were more >20yo patients than <20 yo patients? Wouldn’t that be expected based on population and need distributions?

29. These conclusions don’t seem to arise from the results.

49. Sp ‘Jordon’

70. The point concerning rehab professionals being included in acute care probably needs a citation. I am not convinced this was a universal experience.

71. Citations 4,5 don’t argue that rehabilitation services have been provided in a medical model. What point is being made here? Is it that medical rehabilitation services are dominant in Jordan? If so, it is reasonable for the authors to assert that – ideally – but not necessarily – with a citation or two. As opposed to what, though? A mix of community rehabilitation, or patient centred rehabilitation provided in health services? Why is the point necessary at all?

About lines 80-120. There are one or two main points here. They could be shortened. The points seem to be that rehabilitation is a) often time-bound and b) was threatened or reduced during covid. That’s all that needs to be said. The lengthy but ultimately fairly vague description of the value of rehabilitation in general and in subacute care isn’t necessary.

About here, provide more detail about lockdowns – what exceptions were there? Was rehabilitation a permissible reason to move in the community, or not. If so, the downward trend would align with consumer choice rather than compliance with health measures.

149. Data based? Or Database

154. Isn’t time or time epoch or ‘policy setting’ an IV?

157. Provide a rationale for using number of individual attendees rather than episodes or care, visits, attendances or other variable.

193. Figure 1 needs to be improved. Tidy the vertical lines and make it clear what they represent. The ‘period’ headings are crude text boxes. The x axis is month (or date), not time. T axis is total patients per month.

195. This analysis leaves room for concern. Comparing the slope compares the rate of change during the periods, not the total number of patients seen between the periods. That’s a valid comparison (restrictions reduce presentation) but doesn’t strictly answer the question whether more or less people present

210. If there is a conclusion that women did better (slowed their presentation less than men) great caution is needed to control for other factors. Maybe women were richer/poorer, greater/lesser need; younger/older; lived further awa/closer etc. State explicitly whether or not this comparison controlled for other factors. If it doesn’t, it’s not a reasonable conclusion.

214. This graph also needs work. Make the dimensions the same as figure 1. Add a key for women. Figures 2 duplicates figure 1. Why couldn’t M, F and total appear on the same chart?

216. While the slopes for men and women might be statistically different, what is the line really saying. To me, it says that men and women both slowed to 0 presentations by the hard cut-off date. The women necessarily slowed faster because they presented more frequently before the pandemic started. To assert women did better is quite meaningless in my view. This is my concern with the analysis overall. I am uncertain whether a) differences were gender-related without knowledge about controlling for other factors and b) I think your focus on the statistics relative to the lines themselves is detracting from the main point. Specifically, public health directives slowed presentations to rehabilitation for all, with modest differences between women and men – but ultimately the outcome was the same: rehabilitation slowed to nothing, and returned after relaxation of measures.

For all these analyses, historical summaries of presentations would be helpful. To what extent are these data just cyclical? Would a comparison with the previous years on average (rather than assessing the slope) be a better method?

 230. Again, review the chart. To me, the data suggest children returned to usual numbers while adults increased, rather than saying children fared more poorly than adults. In any case, and this is another concern with the paper overall – how much of these fluctuations are seasonal or just random variance?

237. What are we learning in this analysis? Is it that reduced attendance was independent of condition? If so, say so – no chart is needed.

290. It isn’t reasonable to conclude that children have less access. Fewer children presented – but that might be quite reasonable in terms of a) more adults needed care due to more needs in the population b) a greater proportion of need was met in children (but we have no data about that here).

295. We simply cannot conclude that services for children are inadequate on the basis of these data. I do not doubt there are real issues for pediatric rehabilitation – but this study really does not tell us anything about that. That there were fewer services tells us nothing about the adequacy of those services relative to the need. This cannot be part of your results, discussion or conclusion. It can however be a limitation – you can argue that your work does not answer anything about unmet need relative to the need, just the raw numbers of two broad age ranges.

309. Be clear that these points of discussion are not directly arising from the results of the study. For example, you could say ‘while there is limited evidence about the relative met need of different groups, the overall decline in services in medical settings is a concern, and could be addressed with family-centred (family-led?) approaches. However my concern there is that there is no reason to conclude those services would be any less susceptible to public health measures than hospital based services. You will need to scale back the emphasis on this conclusion, however valid it might be based on your expertise: it simply doesn’t follow from your findings in this study.

320. Again, I don’t see how your findings compel a paradigm shift about much at all, never mind a ‘top down’ model, which is neither compelling or clear.

333. I have no doubt these policy implications are sound. However, it isn’t reasonable to imply your findings are direct evidence for the merit of an existing strategy. This appears to be – and is – an over-reach. It is adequate to say ‘how rehabilitation can be resilient to health systems shocks needs to be a consideration in the implementation of new strategies’ or similar

Conclusion. The conclusion is too long. I think your conclusion is basically that Covid public health measures reduced access to rehabilitation. New approaches to build resilience and access to rehabilitation during public health emergencies are needed. There are challenging questions about how current strategies for rehabilitation in Jordan's health system can include strategies for consumer led (or similar phrase?) strategies that might mitigate against major disruptions to health services and provide more client-focused rehabilitation.

Check citation 9 author spelling O’’’’Dell

Author Response

Authors’ response to reviewer 3

Thank you for your comments which improved our paper. Changes are highlighted in yellow in the attached file. Below are the responses to your comments.

Comment 1:

Middle and low resource settings is an atypical phrase. Is middle and low income setting or country better? This matters in terms of both consistency and for future searches.

Author’s response:  we changed the term to middle- and low-income countries. The consistency of the term used was checked throughout the paper.

Comment 2:

Why are policies and procedures ‘meso’. Isn’t policy macro, while procedures might be a mix of levels? Why is the term needed at all?

Authors’ response: Based on the reference: Negrini, S., Mills, J., Arienti, C., Kiekens, C., & Cieza, A. (2021). “Rehabilitation research framework for patients with COVID-19” defined by Cochrane rehabilitation and the World Health Organization rehabilitation program. Archives of Physical Medicine and Rehabilitation, 102(7), 1424-1430. https://doi.org/10.1016/j.apmr.2021.02.018

A need for studies that examine the effect of COVID-19 on meso and macro systems are needed. Therefore, we used this to support the aim of the study. You are right policies are macro systems. So, we removed the terms meso and macro from the aims as they are not needed. We kept them through the paper in discussion to highlight how our study fills the gap in research. We also checked that macro stands for policies while meso reflects access to services.

Comment 3:  

Delete full stop after patients

Authors’ response: Deleted.

Comment 4:

Is this analysis age corrected? Why is it important that there were more >20yo patients than <20 yo patients? Wouldn’t that be expected based on population and need distributions?

Authors’ response:  The analysis is not age-corrected, the first aim was to describe the overall trend, and because we used total numbers of patients this type of analysis doesn’t require age adjustment. The second aim includes comparisons between 2 groups based on gender and Age.

The importance is examining the trend of change in children and young adults populations versus adults and elderly. As a sole provider of public rehabilitation services, we wanted to examine who was affected more during the lockdown, whether was it children and youth or adults and elderly. The Jordanian population's median age is 23.8 years. Public health records in the ministry of health were divided into>20yo patients and <20 yo patients so we kept the same categorization to provide relevant information to policymakers.

 Comment 5:

These conclusions don’t seem to arise from the results.

Authors’ response: We revised the conclusions as follows “Rehabilitation services in Jordan were reduced as a result of COVID-19 public measures. New approaches to building resilience and access to rehabilitation during public health emergencies are needed. It is necessary to develop a new model of rehabilitation care in order to mitigate major disruptions to health services and to provide a more client-focused approach to rehabilitation.”.

Comment 6

  1. Sp ‘Jordon’

Authors’ response:  corrected at 2 places in the txt. Thanks for pointing this out.

Comment7

  1. The point concerning rehab professionals being included in acute care probably needs a citation. I am not convinced this was a universal experience.

Authors’ response:  The citation is added it is the study of Prvu Bettger, J., Thoumi, A., Marquevich, V., De Groote, W., Rizzo Battistella, L., Imamura, M., Delgado Ramos, V., Wang, N., Dreinhoefer, K. E., Mangar, A., Ghandi, D. B., Ng, Y. S., Lee, K. H., Tan Wei Ming, J., Pua, Y. H., Inzitari, M., Mmbaga, B. T., Shayo, M. J., Brown, D. A., … Stein, J. (2020). COVID-19: Maintaining essential rehabilitation services across the care continuum. BMJ Global Health, 5(5), e002670. https://doi.org/10.1136/bmjgh-2020-002670

Comment8

  1. Citations 4,5 don’t argue that rehabilitation services have been provided in a medical model. What point is being made here? Is it that medical rehabilitation services are dominant in Jordan? If so, it is reasonable for the authors to assert that – ideally – but not necessarily – with a citation or two. As opposed to what, though? A mix of community rehabilitation, or patient centred rehabilitation provided in health services? Why is the point necessary at all?

Authors’ response:  Thank you for your comment. The point we are trying to make here is that Rehabilitation in Jordan is practiced within a traditional medical model of care* (Farre, A., & Rapley, T. (2017). The New Old (and Old New) Medical Model: Four Decades Navigating the Biomedical and Psychosocial Understandings of Health and Illness. Healthcare (Basel, Switzerland)5(4), 88. https://doi.org/10.3390/healthcare5040088)

 which is restricted to physical medicine specialist diagnosing patients in clinics and referring them to therapist and mostly physiotherapist to receive modalities treatment. This is the situation that the two reference describe in Jordan. Consequently, rehabilitation is not accessible or affordable in Jordan. In addition, rehabilitation is recognized as tertiary care which is unnecessary services in emergencies.

In this paragraph, we are providing background information about the rehabilitation situation in Jordan to give the reader context to appreciate the implications and recommendations. We hope this answers your questions. We will be happy to address it further if you provide specific suggestions.

Comment 9

About lines 80-120. There are one or two main points here. They could be shortened. The points seem to be that rehabilitation is a) often time-bound and b) was threatened or reduced during covid. That’s all that needs to be said. The lengthy but ultimately fairly vague description of the value of rehabilitation in general and in subacute care isn’t necessary.

About here, provide more detail about lockdowns – what exceptions were there? Was rehabilitation a permissible reason to move in the community, or not. If so, the downward trend would align with consumer choice rather than compliance with health measures.

Authors’ response:

This section of the paper was extensively edited to respond to your suggestions as well as the suggestions of the other two reviewers. The text has been rearranged as recommended and additional information added to clarify what was and was not affected by government policies and responses to COVID.

Comment 10

  1. Data based? Or Database

Authors’ response:  It is a written Data base.

Comment11

  1. Isn’t time or time epoch or ‘policy setting’ an IV?

Authors’ response:  Yes, we added that now to the IV list.  “three time periods based on governmental policies imposed,”

Comment 12

  1. Provide a rationale for using number of individual attendees rather than episodes or care, visits, attendances or other variable.

Authors’ response:  We meant to say, " the number of patients who visited the rehabilitation clinics during each period of time” Which is the same as episodes of care, or a number of visits.

We rewrote the sentence “In this study, the number of patients who visited the rehabilitation clinic during each period of time was examined” and hope it’s clear now.

Comment 13

  1. Figure 1 needs to be improved. Tidy the vertical lines and make it clear what they represent. The ‘period’ headings are crude text boxes. The x axis is month (or date), not time. T axis is total patients per month.

Authors’ response:  We modified the figure as noted. We hope that the period is now clearly represented by the boxes.

Comment 14

  1. This analysis leaves room for concern. Comparing the slope compares the rate of change during the periods, not the total number of patients seen between the periods. That’s a valid comparison (restrictions reduce presentation) but doesn’t strictly answer the question whether more or less people present

Authors’ response:

The trendlines (slopes) for time-dependent data which is the case in our study are measures of the “systematic increase or decrease in the Y-axis measures and thus represent the slope over time (where the time units are whatever time periods separate the repeated measures)” (Cohen et al., 2003, p. 576). The slopes, therefore “tell us” how many patients were seen per month during the three phases of the study as shown on the Y-axes of the graphs.

Reference: Cohen, J., Cohen, P., West, S. G., & Aiken, L. S. (2003). Applied multiple regression/correlation analysis for the behavioral sciences (3rd ed.). Lawrence Erlbaum Associates Publishers.

Comment 15

  1. If there is a conclusion that women did better (slowed their presentation less than men) great caution is needed to control for other factors. Maybe women were richer/poorer, greater/lesser need; younger/older; lived further awa/closer, etc. State explicitly whether or not this comparison controlled for other factors. If it doesn’t, it’s not a reasonable conclusion.

Authors’ response: we agree with your comment. Therefore, we added this sentence to the limitation of the study “Considering that we were unable to access socioeconomic variables in the HAKEEM system, we were not able to control for factors such as income, education, and residency when examining the change in the number of patients who visited the rehabilitation clinics."

Comment 16

  1. This graph also needs work. Make the dimensions the same as figure 1. Add a key for women. Figures 2 duplicates figure 1. Why couldn’t M, F and total appear on the same chart?

Authors’ response:  We formatted all the figures for size and labeling consistency. We kept figures 1, 2,3 separate to decrease the crowdedness of figures and to make it clear that we ran 3 separate comparisons (Time series analyses).

Comment 17

  1. While the slopes for men and women might be statistically different, what is the line really saying? To me, it says that men and women both slowed to 0 presentations by the hard cut-off date. The women necessarily slowed faster because they presented more frequently before the pandemic started. To assert women did better is quite meaningless in my view. This is my concern with the analysis overall. I am uncertain whether a) differences were gender-related without knowledge about controlling for other factors and b) I think your focus on the statistics relative to the lines themselves is detracting from the main point. Specifically, public health directives slowed presentations to rehabilitation for all, with modest differences between women and men – but ultimately the outcome was the same: rehabilitation slowed to nothing, and returned after the relaxation of measures.

Authors’ response:  We agree with the reviewer’s comment. We changed better access to be more accessible and we added this sentence afterward to address the comment.  “However, without controlling for socioeconomic variables, it is impossible to establish that this difference is gender-related.”  In addition to what we mentioned before that we added this to study limitations, we hope that this addresses the reviewer's comment.

Comment 18

For all these analyses, historical summaries of presentations would be helpful. To what extent are these data just cyclical? Would a comparison with the previous years on average (rather than assessing the slope) be a better method?

Authors’ response:  We thought about this before, but unfortunately obtaining data from the Ministry records was not an easy thing. We placed a request to obtain the data for the years before the COVID-19 pandemic but we were not able to do so. For that, we have used time series analysis to compare the data over certain periods and compare it within these time frames.

Comment 19

  1. Again, review the chart. To me, the data suggest children returned to usual numbers while adults increased, rather than saying children fared more poorly than adults. In any case, and this is another concern with the paper overall – how much of these fluctuations are seasonal or just random variance?

Authors’ response:  We agree with this comment. This is why we are just stating facts related to the periods we are examining, we are not extrapolating our findings to times other than the three periods of time we are examining. The results are written to support this point. We are not comparing the COVID-19 period with the time before, we are comparing three times were different governmental procedures are imposed. We hope this explanation responds to your concern.

Comment 20

  1. What are we learning in this analysis? Is it that reduced attendance was independent of the condition? If so, say so – no chart is needed.

Author’s response:  In this analysis, we meant to show the number of patients who attend different Rehabilitation clinics during the study timeframe. The figure helps explain which clinics were mostly visited by patients. We are happy to remove it if it remains a concern for reviewer. 

Comment 21

  1. It isn’t reasonable to conclude that children have less access. Fewer children presented – but that might be quite reasonable in terms of a) more adults needed care due to more needs in the population b) a greater proportion of need was met in children (but we have no data about that here).

Authors’ response: We agree that both interpretations are possible and as you mentioned we will not be able to prove that from the data we already analyzed. But the observed trend is significant and can be explained both ways. Therefore, we rewrote this part as follows to address this comment  “During the pandemic, however, adult and elderly patients were more likely to require rehabilitation services, which can be attributed to the increased need for these services. During the pandemic, children have limited access to rehabilitation services, as compared to adults and the elderly. Considering that most children who receive rehabilitation services in the public sector in Jordan have disabilities[8], there might be a need to consider the inclusion of children with disabilities in COVID-19 in national response policies in Jordan[15].  ”

Comment 22

  1. We simply cannot conclude that services for children are inadequate on the basis of these data. I do not doubt there are real issues for pediatric rehabilitation – but this study does not tell us anything about that. That there were fewer services tells us nothing about the adequacy of those services relative to the need. This cannot be part of your results, discussion, or conclusion. It can however be a limitation – you can argue that your work does not answer anything about unmet needs relative to the need, just the raw numbers of two broad age ranges.

Authors’ response:  We echo this point. The sentence was rewritten as follows “Considering that most children who receive rehabilitation services in the public sector in Jordan have disabilities5, there might be a need to consider the inclusion of children with disabilities in COVID-19 in national response policies in Jordan”

Comment 23

  1. Be clear that these points of discussion are not directly arising from the results of the study. For example, you could say ‘while there is limited evidence about the relative met need of different groups, the overall decline in services in medical settings is a concern, and could be addressed with family-centred (family-led?) approaches. However my concern there is that there is no reason to conclude those services would be any less susceptible to public health measures than hospital-based services. You will need to scale back the emphasis on this conclusion, however valid it might be based on your expertise: it simply doesn’t follow from your findings in this study.

Authors’ response:  This part is revised to respond to the comment as following “Although the results of this study provide limited information regarding how children with disabilities were met during the pandemic, the overall decline in rehabilitation services within medical settings is concerning. It is suggested that a biopsychosocial model be adopted wherein the role of the family in providing care for their children is enhanced through family-centered and collaborative models of rehabilitation, particularly for follow-up and continuity of care”

Comment 24

  1. Again, I don’t see how your findings compel a paradigm shift about much at all, never mind a ‘top down’ model, which is neither compelling nor clear.

Authors’ response: Based on our experience with medical rehabilitation in Jordan and considering the situation of rehabilitation services during COVID-19 and the limited role that rehabilitation professionals played in the national response (needless to say we had no role). We really think this can be a start of paradigm shift for our medical model of care. Therefore, we used this terminology. As for the top-down, we agree with you and this is removed now from the paragraph.

Comment 25

  1. I have no doubt these policy implications are sound. However, it isn’t reasonable to imply your findings are direct evidence for the merit of an existing strategy. This appears to be – and is – an over-reach. It is adequate to say ‘how rehabilitation can be resilient to health systems shocks needs to be a consideration in the implementation of new strategies or similar

Authors’ response:  We have addressed this comment by rephrasing the paragraph  “Therefore, based on the results of the study, the following suggestions are made to ensure the implementation of resilient rehabilitation services in Jordan's health systems: (1) Incorporating acute care rehabilitation services into national health care plans as an essential component of quality care, (2) Establishing policies to ensure continuity of rehabilitation during periods of limited access by regulating telerehabilitation, home programs, and in-home virtual reality services [15–20], (3) requiring rehabilitation professionals to attend continuing education courses that address the competencies of the rehabilitation workforce, and (4) Providing services to children with disabilities using a biopsychosocial and family-centered model as best practices, and (5) promoting public-private partnerships among healthcare sectors to meet the increased demand for rehabilitation services.

Comment 26

Conclusion. The conclusion is too long. I think your conclusion is basically that Covid public health measures reduced access to rehabilitation. New approaches to building resilience and access to rehabilitation during public health emergencies are needed. There are challenging questions about how current strategies for rehabilitation in Jordan's health system can include strategies for consumer-led (or similar phrases?) strategies that might mitigate against major disruptions to health services and provide more client-focused rehabilitation.

Authors’ response:  Thank you for your suggestion. We have edited this section as follows. Hope it reads better now. “The COVID-19 pandemic public measures in Jordan reduced access to rehabilitation services. New approaches to building resilience and access to rehabilitation during public health emergencies are needed. Jordan's health system faces challenges related to how current rehabilitation practice models can include both evidence-based and family-centered care models that may help mitigate major disruptions to health services as well as provide more client-focused rehabilitation”.

Comment 27

Check citation 9 author spelling O’’’’Dell

Authors’ response:  the citation is corrected.

Round 2

Reviewer 3 Report

Thank you to the authors for their thoughtful revisions.

The manuscript is much-improved, but I remain concerned about the conclusions and recommendations, which are not connected to the findings. The findings are interesting and important. The authors expertise and judgement is sound – but the writing needs to be clear about what is a) found, b) arises from the findings and c) seems like a useful direction to explore but is not directly related to the findings. General and specific comments follow.

Please use the phrase low and middle-income (LMIC).

You cannot conclude which age group was affected more using this analysis. You will need to delete reference to that comparison, or use a statistic appropriate to make that comparison. I don’t know that it is necessary or useful to make that comparison given your aim. Describe how policy settings affected the different age groups, without necessarily comparing them. That is useful.

I remain uncomfortable with your conclusion. Some minor wordsmithing is all that is needed. Rather than saying ‘it is necessary to…’, a reasonable conclusion seems to be that strategies to avoid the shutdown of rehabilitation in public health ‘should be examined’ without concluding what they should be ‘new models/client focused’. You could, if necessary, argue that ‘possible avenues to explore are client-focused models’, but without elaborating, it isn’t clear how a client focused model solves the problem anyway. Your paper is strong enough without this over-reach.

The figures are improved. Consider whether better descriptors of the ‘periods’ would help. Why not make it clear what happened, when? Note when different measures were invoked, for example. As it is, I am required to read the paragraph on a different phase to figure out what was happening when.

I defer to the authors and editors on the appropriate statistic to compare mean presentations in the different periods.

You still cannot assert that children had less access. Your changes help. You can assert that less children presented overall, but this says nothing about ‘access’ at all. This needs to be addressed in results and comments. It is almost inevitable that children do have additional access challenges, but we simply cannot conclude that based on your analyses. It is equally plausible the met need (proportion of children who need services who get appropriate care) is higher than for adults – indeed I suspect that is the case, but either way we just cannot say with any confidence. I think a point about disability-inclusion in response measures applies to adults, too. The point is a good one.

Again, I don’t think you can recommend a particular model. You can say it is timely to examine the merits of different models, and briefly explain why a biopsychosocial model is, in your view, appealing.

Your recommendations also need to adopt that language. You cant claim public private approaches, family-centred models or telerehabilitation for example – are any better than status quo – but you can (and should) state that these models are appealing and deserve exploration, and are *potential* solutions.
